# Targeted Destruction of S100A4 Inhibits Metastasis of Triple Negative Breast Cancer Cells

**DOI:** 10.3390/biom13071099

**Published:** 2023-07-10

**Authors:** Thamir M. Ismail, Rachel G. Crick, Min Du, Uma Shivkumar, Andrew Carnell, Roger Barraclough, Guozheng Wang, Zhenxing Cheng, Weiping Yu, Angela Platt-Higgins, Gemma Nixon, Philip S. Rudland

**Affiliations:** 1Department of Biochemistry and Systems Biology, University of Liverpool, Liverpool L69 7ZB, UK; tismail@liverpool.ac.uk (T.M.I.); brb@liverpool.ac.uk (R.B.); platthig@liverpool.ac.uk (A.P.-H.); 2Department of Chemistry, University of Liverpool, Liverpool L69 7ZB, UK; rachelcrick@yahoo.co.uk (R.G.C.); umas.shivkumar@wmc.ac.uk (U.S.); acarnell@liverpool.ac.uk (A.C.); 3Department of Clinical Infection, Microbiology and Immunity, University of Liverpool, Liverpool L69 7ZB, UK; siyu@liverpool.ac.uk (M.D.); wangg@liverpool.ac.uk (G.W.); 4Medical School, Southeast University, Nanjing 230032, China; chengzhenxing407@163.com (Z.C.); yuweiping@seu.edu.cn (W.Y.); 5Department of Gastroenterology, First Affiliated Hospital, Anhui Medical University, Hefei 210009, China

**Keywords:** S100A4, PROTAC destruction, inhibition of migration/metastasis, triple negative breast cancer

## Abstract

Most patients who die of cancer do so from its metastasis to other organs. The calcium-binding protein S100A4 can induce cell migration/invasion and metastasis in experimental animals and is overexpressed in most human metastatic cancers. Here, we report that a novel inhibitor of S100A4 can specifically block its increase in cell migration in rat (IC_50_, 46 µM) and human (56 µM) triple negative breast cancer (TNBC) cells without affecting Western-blotted levels of S100A4. The moderately-weak S100A4-inhibitory compound, US-10113 has been chemically attached to thalidomide to stimulate the proteasomal machinery of a cell. This proteolysis targeting chimera (PROTAC) RGC specifically eliminates S100A4 in the rat (IC_50_, 8 nM) and human TNBC (IC_50_, 3.2 nM) cell lines with a near 20,000-fold increase in efficiency over US-10113 at inhibiting cell migration (IC_50_, 1.6 nM and 3.5 nM, respectively). Knockdown of S100A4 in human TNBC cells abolishes this effect. When PROTAC RGC is injected with mouse TNBC cells into syngeneic Balb/c mice, the incidence of experimental lung metastases or local primary tumour invasion and spontaneous lung metastasis is reduced in the 10–100 nM concentration range (Fisher’s Exact test, *p* ≤ 0.024). In conclusion, we have established proof of principle that destructive targeting of S100A4 provides the first realistic chemotherapeutic approach to selectively inhibiting metastasis.

## 1. Introduction

Breast cancer accounts for about one quarter of cancer cases in women (worldwide 2.2 million in 2020) and nearly a third have died (685,000) of this disease [1]. The deadliest type of breast cancer is that without hormonal receptors, the triple negative breast cancer (TNBC) accounting for 10–15% of all breast cancers [2], with three times the risk of patients dying within 5 years [3]. This increased mortality is due to a lack of specificity of the chemotherapeutic agents employed since specific antihormonal agents are ineffective in receptor-free TNBC. Thus, new treatments for TNBC are of high priority to meet this clinical need. Most patients, however, who suffer from breast and other carcinomas die not from the primary tumour but from dissemination or metastasis [4]. Certain tumour cells possess the ability to migrate and invade the neighbouring tissue and eventually metastasise while others do not [5,6]. These results suggest that specific genes/proteins are involved in the metastatic process that are different from those involved in the production of the primary tumour [7]. These proteins have been termed metastasis-inducing proteins, one such example is S100A4 [8], a member of the S100 calcium-binding family [9]. S100A4 is expressed in most human metastatic cancers and is associated with the premature death of patients with different types of carcinomas, including those from the breast [10], oral mucosa, bladder, pancreas, prostate, colorectum, oesophagus, lung, stomach [11], and thyroid glands [12,13,14]. Although S100A4 cannot stimulate cell proliferation or induce tumour formation alone, it can stimulate the metastatic cascade in model fly and rodent systems by interaction with oncogenes such as ras^Val12^ and neu [8,15,16]. Thus, the great advantage of S100A4 over standard oncogenes as a target for anticancer therapy is that it directly induces and maintains metastases [8,15] and its suppression causes no harmful effects in animals [17]. These facts are in contrast to those of standard oncogenes which cannot induce metastasis directly [8,15,16] and whose suppression as proto-oncogenes causes grave deleterious effects in animals [18].

Although there are several ways to target S100 proteins [19,20], only small molecules can enter cancer cells easily [21]. Small molecules, however, have proved rather inefficient as inhibitory agents for S100A4 [22,23,24]. Thus, niclosamide was only about 50% efficient at inhibiting S100A4 mRNA and its biological effects at 1µM, but this dose overlapped considerably with its target-cell nonviability at about 30% [23]. The most widely used inhibitor is pentamidine, an FDA approved antiprotozoal drug [25,26], only active against S100A4 at high concentration above 100 μM, causing considerable side effects [27] due to lack of specificity. In contrast, a specific target of S100A4 is nonmuscle myosin IIA (NMIIA) which, upon binding, triggers direct increases in cell migration and invasion [28], the principal initial steps in the formation of metastases [29]. Here, we report that we have identified a compound and its chemical analogues that can specifically inhibit this interaction in the 10′s of μM concentration range and have chemically coupled it to a cereblon (CRBN)-stimulator thalidomide [30] to form a proteolysis targeting chimera (PROTAC) mediated by the cellular proteasome [31,32]. The bifunctional PROTAC compound has improved efficiency in inhibitory properties associated with metastasis by nearly 20,000-fold in model rat [33] and human TNBC cell lines [34], with minimal side effects. The lead PROTAC compound demonstrates proof of principle that specific destruction of S100A4 can dramatically reduce in vitro properties associated with metastasis, as well as inhibiting the formation and provoking the destruction of metastases in an experimental model [35,36] of TNBC in mice, suggesting that it has potential as an antimetastatic agent in humans. 

## 2. Materials and Methods

### 2.1. Inhibition of Binding of S100A4 to NMIIA 

A panel of 2500 compounds from CRUK were screened at 33 μM, first robotically and then manually, in an IAsy two channel resonant mirror biosensor (Affinity Sensors, Saxon Hill, Cambridge, UK) for inhibition of binding of recombinant human (rh)S100A4 to recombinant fragment of 149-terminal amino acids of human NMIIA (rC-NMIIA) (Appendix A) [37,38]. The rC-NMIIA was immobilised on aminosilane surfaces using BS³ (Perbio, Chester, UK). The rhS100A4 and rC-NMIIA were produced with a pET-15b expression vector in *E. coli* BL21 (DE3) bacterial cells and purified to greater than 95% on SDS-polyacrylamide gels [38].

### 2.2. Chemical Synthesis

Full experimental details can be found in Appendix A and Appendix A [31,32]. In brief, synthesis of the CT070909 analogue US-10113 (compound **3a**) (Figure 1A, B) was performed by acid reflux of the commercially available endo isomer of (3a*R*,4*R*,7*S*,7a*S*)-Tetrahydro-4,7-methanoisobenzofuran-1,3-dione (**1a**) with *o*-toluidine (**2a**). Carboxylic acid US-10113 analogues **3b** and **3c** required for the full PROTAC compounds were synthesised using the same methodology [32,39,40]. Synthesis of the endo PROTAC after Rachel Gemma Crick (RGC)-01-05-18, henceforth shortened to RGC (Figure 1C), was achieved through initial reaction with 2,3,4,5,6-pentafluorophenol (**4**) followed by amide bond formation by reaction with thalidomide attached to its linker 4-aminobutylacetamide (ABA) (**13**) in the presence of 4-dimethylaminopyridine (DMAP). Compound **13** was synthesised using previously established methodology [32,39,40]. Synthesis of the exo isomer iRGC (Figure 1D) was initiated with the exo isomer (3a*R*,4*R*,7*S*,7a*R*)-3a,4,7,7a-Tetrahydro-4,7-methanoisobenzofuran-1,3-dione (**1b**). In this case, **3c** was reacted directly in the amide coupling step with 1-[Bis(dimethylamino)methylene]-1H-1,2,3-triazolo [4,5-b] pyridinium3-oxidhexafluorophosphate (HATU) in *N, N*-Diisopropylethylamine (DIPEA). All compounds synthesised were greater than 94% pure on HPLC.

### 2.3. Cell Culture

The nonmetastatic rat mammary benign-tumour-derived cell line Rama 37 was isolated and characterised as previously described [5]. (RRID: CVCL-T286). The metastatic S100A4 cDNA- [38] and S100P cDNA- [41] transfected Rama 37 cell lines were established from Rama 37 and characterised as previously reported [38,41]. The levels of S100A4 and S100P in their respective transfected cell lines were seen to be greatly elevated over those basal levels in the untransfected parental Rama 37 cells (Appendix A), in agreement with previous reports of their about 6-fold [42] and 5-fold [43] increases, respectively, over the parental Rama 37 cells. They were all cultured in growth medium (GM) (DMEM, 10% (*v*/*v*) fetal calf serum (FCS), 10 ng/mL insulin, 10 ng/mL hydrocortisone, 100 units/mL penicillin, and 100 μg/mL streptomycin) [38,41]. All three cell lines were sourced from Prof P.S. Rudland’s collection. Human TNBC MDA-MB-231 [34,44] and mouse TNBC 4T1 cells [35] were obtained from American Type Culture Collection (ATCC) (Manassas, VA, USA) and from Dr Weiping Yu, Southeast University, PR China, respectively, who obtained it from China Centre for Type Culture Collection (Wuhan, Hubei, China), and both were routinely cultured as above. Levels of S100A4 in aggressively metastatic MDA-MB-231 and 4T1 cells were considerably higher than in nonaggressive human breast cancer MCF-7 cell line [36]. This is shown for MDA-MB-231 cells in Appendix A. Batches of randomised cell stocks were stored frozen and used within 5 passages [5]. Cell lines were routinely tested every 6 months for mycoplasma contamination using MycoAlert Mycoplasma Detection kit (Lonza, Nottingham, UK).

### 2.4. Measurement of Cell Migration/Invasion

Migration of cell lines was measured 24 h after cells were seeded in Boyden chambers on porous polycarbonate membrane inserts (Appendix A) [38]. For invasion assays, the membranes were coated with Matrigel. Each concentration of a compound was assayed in 4 chambers and means normalised for number of cells in no-compound control wells. Each experiment was conducted 3 times and the overall mean ± SE was usually shown as a percentage of the migration of cells with no compound present set at 100%. The migration and invasion rates of S100A4-transfected Rama 37 cells were seen to be greatly increased over those basal rates of the untransfected parental Rama 37 cells (Appendix A), in agreement with previous reports of about a 2-fold increase in migration rates for S100A4-Rama 37 [42] and about a 3-fold increase for S100P-Rama 37 cells [43] over those rates for the parental Rama 37 cells. The migration activity of MDA-MB-231 cells was affected by and dependent on the level of S100A4 [36,45]. Similarly, the migration of mouse 4T1 cell line was mediated by S100A4 [36].

### 2.5. Measurement of Cytotoxicity

Potential cytotoxicity of US-10113 on Rama 37 cells was obtained by growing them for 24 h in GM and then scoring the number of trypsin-EDTA detached cells (Appendix A). Results are mean ± SE of 3 experiments, normalised to that for no inhibitor set at 100 cells. Potential cytotoxicity of the remainder was obtained by growing cells for 24 h in 96 well plates, fixing and staining them with sulphorhodamine B (SRB) (In Vitro Toxicity Assay Tox6, Sigma Aldrich, Saint Louis MO, 63103, USA); the SRB-stained protein was solubilised, and its optical density recorded in arbitrary units (AU) according to the manufacturer’s instructions (see details in Appendix A). Results are mean ± SE of 8 experiments.

### 2.6. Western Blotting and Quantifications

Cells were plated at 2 × 10^5^ per 10 cm diameter petri dish in GM, 8 dishes for 8 different concentrations of compounds, preincubated for 24 h at 37 °C until 70–80% confluent, and then incubated with fresh GM containing compounds for a further 24 h, yielding about 1.5 mg protein per dish. Usually, 10 µg cell extract was then Western blotted from 15% (*w*/*v*) polyacrylamide SDS gels [16]. Ratio of scanned areas under peak band intensities of S100A4 and S100P to that of actin were recorded using image J software (RRID:SCR-003070). Ratio of mean ± SE of 3 independent experiments and a representative blot are usually shown. Sometimes enhancer for the actin blots was used as stated in Appendix A; enhancer was always used for S100A4 and S100P blots but probing for actin and S100A4/P was always undertaken on the same blot.

### 2.7. Depletion of S100A4 in MDA-MB-231 Cells

Duplicate transient knockdown experiments were undertaken with a mixture of siRNAs to S100A4 added for 48, 72, and 96 h to MDA-MB-231 cells according to the manufacturer’s (Horizon, Cambridge, UK) instructions as detailed in Appendix A and either Western blotted for S100A4 with 30 µg of lysate or assayed for migration with 2000 cells per chamber. Permanent knockdown transfectants were produced and selected by infecting MDA-MB-231 cells with lentiviral shRNAs to S100A4 and/or S100P using a fluorescent reporter for shRNA expression (Horizon). The nontargeted scrambled shRNA, shRNA-1 against S100A4, shRNA-2 against S100A4, shRNA against S100P, and a mixture of shRNA-1 against S100A4 and shRNA against S100P produced the cells lines: nontargeted/shRNA, S100A4 shRNA-1, S100A4 shRNA-2, S100P shRNA, and S100A4/P shRNAs, respectively (details in Appendix A). Cell lines were either Western blotted with 30µg lysate or assayed for migration with 5000 cells per chamber. Mean ± SE of three independent experiments is shown.

### 2.8. In Vivo Metastasis Assay

One of the most commonly used models for the assessment of novel anticancer drugs [35] which is dependent on S100A4 for cancer cell migration and metastasis in syngeneic mice is the TNBC mouse cell line 4T1 [36]. This cell line was used for experimental and spontaneous metastasis assays in syngeneic Balb/c mice, principally as described previously in rats [38]. Ten thousand cells were injected intravenously (iv) into tail veins, or 7500 cells were orthotopically introduced into mammary fat pads of randomised female 4-week-old mice and the compounds were injected subcutaneously (sc) on the same day and then every 2 to 3 days for 3 or 4 weeks, respectively. Minimum numbers of mice needed were determined from power calculations using a previously established incidence of 90% for experimental and 75% for spontaneous metastasis with RGC reducing it to 10%. For alpha = 0.05 and a power of 80%, the minimum numbers to obtain a significant difference were 5 and 8, respectively. Animal groups were coded, and results obtained blind; codes were only broken at the end. Mice were monitored twice weekly for appearance of tumours and culled for autopsy at end of the experimental period or earlier upon first signs of distress or when tumour > 1 cm in diameter. Mice were purchased from Beijing Vital River Laboratory Animal Technology and housed in sterile conditions with unrestricted access to water and food at the Research Centre of Genetically Modified Mice, Southeast University, Nanjing, China. All procedures were performed according to State laws under License (Jiangsu Province 2151981) to Dr Z Cheng and animals were monitored by local inspectors.

Primary tumours, lungs, spleen, colon, and other organs were fixed in 10% (*v*/*v*) formalin and embedded in paraffin wax. Sections of 4 μm were cut and stained with haematoxylin and eosin [5] or with antibodies to S100A4 (Thermo Scientific) as described in Appendix A [46,47]. Six microscopic fields from 2 sections from each tumour/tissue were examined by 2 independent observers at 200× magnification (0.68 mm^2^ field) with a minimum of 200 cells per field. Mice with visible lesions > 20 malignant cells in the lung were scored as positive for metastasis. Mice with blocks of striated muscle infiltrated with malignant cells at the edge of the primary tumour were scored as positive for invasion; any mice with tumour sections containing no bordering skeletal muscle were omitted. Immunohistochemical staining of parenchymal tissue cells (spleen, colon) [48] or carcinoma tumour cells was expressed as a mean percentage ± standard deviation (SD), where n= number of mice [16].

### 2.9. Data Analysis

The concentration of inhibitor necessary for 50% inhibition (IC50%) ± SE was determined by the AAT Bioquest IC50 calculator (https://www.aatbio.com/tools/ic50-calculator) accessed on 9 January 2022 using data from Excel. Calculator gives equation of 4 parameter logistic curve, the corresponding graph, and the IC_50_ ± SE for 4–8 concentrations used. For continuous variables, probability of the difference between two groups (*p*)*,* usually between the zero and a given concentration of inhibitor, was calculated using Student’s *t*-test (n, number of independent experiments, usually = 3) and the probability of difference in percentage staining between different sets of tissues/tumours was calculated from one way analysis of variance (ANOVA) (n = 4 to 8), all supplied with software from Stats Direct Ltd. (Cambridge, UK). For categorical variables, probability of difference in incidence of invasion or metastasis between two groups, usually between zero and a given concentration of inhibitor, was calculated using Fisher’s Exact test (n = 5 to 8) with SPSS software version 22 (RRID:SCR-002865) (SPSS, Chicago, IL, USA). All statistical tests were 2-sided, and differences were considered significant when *p* < 0.05.

## 3. Results

### 3.1. Identification of Inhibitor of S100A4 Binding to NMIIA and Migration of Rat Mammary Cells

A library of 2400 compounds from Cancer Research UK (CRUK) was screened for the ability to inhibit binding of S100A4 to NMIIA at 33 μM using robotic surface plasmon resonance (SPR) [37]. This process yielded 33 hits of compounds with relative binding reduced to 50% or below (Appendix A). This step was followed by manual screening which yielded a single hit CT070909 (Figure 1A) with over 90% inhibition of binding at 33 μM (Appendix A). The chloride atoms rendered the compound relatively insoluble and a structurally simpler compound US-10113 was synthesised in house (Figure 1B, Appendix A). Both compounds significantly reduced the migration of S100A4-transfected rat mammary (Rama) 37 cells to a similar 33% (Student’s *t* test, *p* = 0.91) of the original value at 33 μM (*p* = 0.001 for US-10113 and *p* = 0.002 for CT070909), whereas the parental S100A4-negative Rama 37 cells showed no significant reduction with either compound (*p* = 0.27 and 0.56, respectively) (Appendix A). US-10113 inhibited the migration of S100A4-transfected Rama 37 cells in a dose-dependent manner with 95% inhibition at 100 μM and IC_50_ (mean ± SE) of 46 ± 4 μM with no significant toxicity up to 500 μM (*p* = 0.15) (Figure 2A–E*).*

### 3.2. PROTAC Modification of Inhibitor in Rat Cell Lines

To improve its efficiency, inhibitor US-10113 was synthesised *de novo* coupled to thalidomide (Figure 1C), as shown in Appendix A to yield PROTAC RGC-01-05-18 (RGC). There was no significant toxic effect of RGC on parental Rama 37 cells up to 100 μM (Student’s *t*-test, *p* = 0.58) (Figure 2B). However, RGC reduced the level of S100A4 by 99% at 100 nM in S100A4–transfected Rama 37 cells with an IC_50_ of 38 ± 2 nM (Figure 2C), whereas there was no significant reduction in the closely-related S100P up to 1000 nM (*p* = 0.19) in the S100P-transfected Rama 37 cells (Figure 2D) (Appendix A). Moreover, RGC reduced the migration of S100A4-transfected Rama 37 cells by 80% at 10 nM with an IC_50_ of 1.6 ± 0.2 nM (Figure 2F) and their invasion by 85% at 100 nM with an IC_50_ of 12 ± 1 nM (Figure 2G). There was no significant reduction in migration of S100P-transfected Rama 37 cells up to 1000 nM RGC (Figure 2H) or thalidomide alone (Figure 2I) (Student’s *t*-test, *p* = 0.19 or 0.34, respectively). There was, however, a significant reduction of 60% in migration of S100P-transfected Rama 37 cells at 2000 nM RGC (*p* = 0.008) (Figure 2H).

### 3.3. Effect of Inhibitor and PROTAC on Human TNBC Cell Line

The addition of inhibitor US-10113 and PROTAC RGC showed no toxicity for human TNBC cell line MDA-MB-231 up to 200 μM and 1000 nM, respectively, (Student’s *t* test, *p* = 0.078 and 0.16) (Figure 3A,B), suggesting little cell death and apoptosis. There was also no significant lowering of S100A4 or S100P up to 100 μM of US-10113 (Student’s *t*-test, *p* = 0.49 or 0.28, respectively), but at 200 μM there was an 83% inhibition of S100A4 (*p* = 0.0001) without a corresponding decrease in S100P (*p* = 0.054) (Figure 3C) (Appendix A). However, the PROTAC RGC reduced the S100A4 level by 74% at 10 nM with an IC_50_ of 3.2 ± 1.0 nM (Figure 3D) (Appendix A). The band intensities at 10–500 nM RGC were very low and fluctuating; some of these small fluctuations were seen in individual blots (e.g., 50 nM RGC: Figure 3D). In contrast, the enantiomer of the normal endo form of the pentane ring of RGC (Figure 1D), the exo isomer RGC (iRGC), failed to reduce significantly the S100A4 level up to 50 μM (*p* = 0.37), thereafter reducing it by 49% at 100 μM (*p* = 0.017) (Figure 3E). There was also little effect of RGC on S100P up to 500 nM (*p* = 0.083), but a reduction of 39% at 1000 nM (*p* = 0.002) (Figure 3F). The inhibitor US-10113 reduced the migration of MDA-MB-231 cells by 96% at 100 μM with an IC_50_ of 56 ± 6 μM (Figure 3G), whereas PROTAC RGC reduced migration by 92% at 10 nM with an IC_50_ of 3.5 ± 1.4 nM (Figure 3H). This result can be seen in the reduced number of cells migrating through the Boyden chamber membrane at 10 nM and 20 nM compared to that without the addition of RGC (Appendix A). There was no significant effect of thalidomide (*p* = 0.95) (Figure 3I) or iRGC (*p* = 0.98) (Figure 3J) up to 2 μM or 100 μM, respectively.

### 3.4. Knockdown of S100A4 in Human TNBC Cells

When MDA-MB-231 cells were transiently transfected with short interfering(si) RNA against S100A4 (Materials and Methods), they produced a decline in S100A4 (Figure 4A) (Appendix A) and in cell migration (Figure 4B) with time, reaching minimum values, respectively, of 9.3% (*p* = 0.0002) and 29.2% (*p* = 0.006) of the initial level after 96 h, with half minimal at 33 h in each case. When MDA-MB-231 cells were permanently transduced with two separate short-hairpin (sh) lentiviral vectors against S100A4, A4 shRNA-1, A4 shRNA-2 and a mixture of shRNAs to S100A4 and S100P, A4/S100P shRNAs (Materials and Methods), they reduced levels of S100A4 to ≤6% of the value obtained with control nontarget shRNA (Figure 4C) (Appendix A). The migration of the relevantly transduced MDA-MB-231 cells was also significantly reduced to 24% (*p* = 0.004), 10% (*p* = 0.002) and 2% (*p* = 0.001), respectively (Figure 4D) (Appendix A). When increasing concentrations of RGC were added to the nontarget shRNA-transduced MDA-MB-231 cells, there was a significant reduction in migration to 7% of the original value at 100 nM (*p* = 0.0008) with an IC_50_ of 15 ± 1 nM (Figure 4E), whereas there was little reduction with the A4 shRNA-1 or A4 shRNA-2 cells (Figure 4F,G) (Appendix A). In contrast, when MDA-MB-231 cells were knocked down by lentiviral vector to S100P, cell migration was significantly reduced to 29% at 100 nM RGC (*p* = 0.0003) with an IC_50_ of 18 ± 3 nM (Figure 4H), but the doubly transduced A4/S100P cells showed no significant effect (*p* = 0.64) (Figure 4I) (Appendix A).

### 3.5. Effect of PROTAC on Metastasis in TNBC Mouse Model

When TNBC 4T1 mouse mammary cells [35] were injected with RGC at 7 μg/kg to 3.5 mg/kg (about 10 nM to 5 μM), either iv or orthotopically into syngeneic Balb/c mice and resultant lesions were analysed (Appendix A, they produced, respectively, 100% of mice with experimental lung metastasis or 100% with locally invasive primary tumours and spontaneous lung metastases (Appendix A) (Table 1). When RGC was injected simultaneously with the cells and then every 2 to 3 days thereafter, the percentage of immunohistochemically stained cells for S100A4 in the spleen and colon dropped significantly by 2–4 fold depending on the dose (Student’s *t*-test, *p* ≤ 0.026), but there was no significant decrease for 13.5 mg/kg US-10113 or 1.35 mg/kg thalidomide (*p* ≥ 0.34) (Appendix A) (Table 2). The incidence of experimental metastasis was also significantly reduced by RGC to 20% for 0.07 and 0.70 mg/kg (Fisher’s Exact test, *p* = 0.024), slightly rising to 40% at 3.5 mg/kg (*p* = 0.083); there was no significant effect for US-10113 or thalidomide (*p* ≥ 0.22) (Appendix A). Similarly, all doses of RGC (0.007 to 3.5 mg/kg) significantly reduced the incidence of local invasion in the primary tumour (*p* ≤ 0.026) (Appendix A) and in spontaneously occurring lung metastases (*p* ≤ 0.038) (Appendix A) (Table 1). Surprisingly, the highest 3.5 mg/kg dose of RGC produced a nonsignificant increase in the incidence of mice with experimental or spontaneous metastasis over lower doses (*p* ≥ 0.24) (Table 1). Moreover, although there was a significant fall in percent primary tumour cells stained for S100A4 after injection of RGC (Student’s *t*-test, *p* ≤ 0.026), the few tumours that grew as experimental lung metastases showed no significant reduction (*p* ≥ 0.32) (Appendix A) (Table 2). To investigate the subcellular location of this stain for S100A4, sections were viewed at higher power in the microscope. The percentage of tumour cells with stained nuclei significantly increased by more than 3-fold in mice injected with RGC (ANOVA Test, *p* ˂ 0.00001), whilst normal cells in spleens and lungs did not (Appendix A, Appendix A). When RGC was injected after establishment of the experimental lung metastases for 14 days, there was a significant drop in the number of lung metastases (Student’s *t*-test, *p* = 0.043) and falls of borderline significance in lung incidence (Fisher’s Exact test, *p* = 0.096), organs involved (Student’s *t*-test, *p* = 0.20) and total lesions (*p* = 0.088) per mouse at 0.07 mg/kg; thalidomide provoked no significant effects (*p* ≥ 0.84) (Appendix A, Appendix A).

## 4. Discussion

The rat and human cell systems used here have been validated previously for the causative link between S100A4 and cell migration/invasion [33,34,36,38,42,45,46,47]. Thus, the levels of S100A4 and S100P are increased after transfection of benign rat Rama 37 cells with their respective cDNAs (Appendix A), and human MDA-MB-231 contains appreciably more S100A4 than less aggressive MCF-7 cells (Appendix A). Moreover, the S100A4/P transfected rat cell lines migrate/invade faster than the untransfected Rama 37 cells (Appendix A) and are shown quantitatively for cell migration in Appendix A. Furthermore, depletion of S100A4 in MDA-MB-231 cells considerably reduces their migration rate (Appendix A) showing their strong dependency on S100A4 for cell migration. The mouse cell line 4T1 is routinely used for testing therapeutic agents against metastatic TNBC in syngeneic mice in vivo [35], and its cell migration in vitro and metastasis in vivo is mediated by S100A4 [36]. Using these model systems, we have shown for the first time that a metastasis-inducing protein S100A4 which is widely expressed in most human cancers [12,13,14] can be inhibited much more efficiently by converting a chemical inhibitor to a specific proteolysing agent inside the cancer cell. The inhibitory compound US-10113′s efficiency at inhibiting cell migration as measured by IC_50_ is similar in rat and TRN breast cancer cell lines at 46 ± 4 and 56 ± 6 µM, respectively (Figure 2E and Figure 3G), whereas the same compound armed with thalidomide, a PROTAC stimulator of natural proteolysis [31,32], yields IC_50_ of 1.6 ± 0.2 and 3.5 ± 1.4 nM, respectively (Figure 2F and Figure 3H), an improvement of nearly 20,000-fold. This reaction of RGC for S100A4 is specific for both the structure of the S100 molecule and that of RGC, since neither the structurally closely related S100P [49] nor the isomeric exo form iRGC are active in the same concentration range (Figure 2D and Figure 3E,J) (Appendix A). However, a higher concentration of 2 µM RGC produces a 60% reduction in cell migration of S100P-transfected rat breast cells (Figure 2H) suggesting that RGC is about 1000 times less effective at inhibiting S100P-transfected than S100A4-transfected Rama 37 cells. It is surprising that US-10113 has no effect on migration of untransfected rat cells at 33 µM (Appendix A), whereas US-10113 can suppress the migration of the S100A4-transfected rat cells almost completely at 75 µM (Figure 2E). However, RGC almost completely abolished the migration of S100A4 transformed rat cells at 10 nM but failed to inhibit the most closely related S100P-transformed rat cells at 1000 nM (Figure 2F,H), clearly delineating the specific effect of S100A4. Since there was no decrease in cell migration when thalidomide was added to S100A4-transfected rat [Figure 2I] or human MDA-MB-231 cells [Figure 3I], it is highly unlikely that thalidomide itself would be toxic at these concentrations.

The fact that RGC’s IC_50_ for the degradation of S100A4 in and the migration of the human breast cancer cell line MDA-MB-231 are similar at 3.2 and 3.5 nM, respectively (Figure 3D,H) (Appendix A) suggests that RGC is operating through a common mechanism. Since there is no degradation of S100A4 with US-10113 up to 100 µM (Figure 3C) (Appendix A), the increased biological activity of RGC over US-10113 may be attributed to its ability to stimulate intracellular proteolytic digestion of S100A4 (Appendix A) and not of other molecules including actin (Figure 3D) or S100P at the same concentration (Figure 3F). Since assays for migration and invasion behave similarly in the S100A4-transfected Rama 37 cells with RGC, there was no need to repeat the invasion assay with the human TNBC cell line. The reason for the low fluctuating levels of S100A4 above the IC_50_ at 10–500 nM is unknown. The inability of similar concentrations of iRGC to stimulate proteolytic degradation of S100A4 and to inhibit cell migration (Figure 3E,J) further supports this conjecture. That degradation of S100A4 causes the decrease in MDA-MB-231 cell migration rates produced by RGC is shown in a two-step experiment. Firstly, both siRNA and shRNAs directed against S100A4 produce a significant reduction in migration rates (Figure 4A–D) (Appendix A). Secondly, RGC causes a significant reduction in migration rates only in control nontarget shRNA- and S100P shRNA-transduced cells and not in S100A4 shRNA-transduced cells in which S100A4 is virtually absent (Figure 4E–H). The fact that transduction of MDA-MB-231 cells by S100A4 shRNAs reduces migration rates of the subsequent transfectants to 10–24%, but that further transduction by S100P shRNA reduces it to 2% (Figure 4D), suggests that S100P is responsible for a small component of the overall migration rate in these cells. This suggestion is confirmed by the loss of the ability of RGC to inhibit migration rates of the doubly transduced S100A4/P cells (Figure 4I). Previously, S100A4/P has been shown to bind preferentially to NMIIA leading to a redistribution of the cytoskeleton, loss of focal adhesions and thereby allowing unopposed locomotory actions of the unaffected NMIIB filaments to drive the cells forward [20,28,50]. Proteolytic destruction of S100A4 is therefore probably a reversal of this effect, leading to decreased migration, invasion, and metastasis [29].

Results in vivo with the mouse cell line 4T1 which was isolated from a spontaneously metastatic mouse TNBC tumour, and which depends on the expression of S100A4 for its metastatic potential [35,36] are largely consistent with those obtained in vitro with the rat and human breast cancer cell lines. Thus, 7µg/kg, corresponding to about 10 nM RGC, suppresses completely the incidence of local invasion into muscle and spontaneous metastasis from the primary tumour to the lungs (Appendix A) (Table 1). However, it is surprising that increasing concentrations of RGC above this value inhibit the incidence of invasion/metastasis with decreasing effect (Table 1); a result which is largely mirrored by the immunohistochemical staining for S100A4 in the spleen, colon and primary breast tumours (Appendix A) (Table 2). Some PROTAC compounds have been reported to exhibit a window of concentrations in which they act as degraders due to competition between the formation of the key 1:1:1 ternary complex and the saturation of the binding sites for two different proteins as separate 1:1 binary complexes; this has been termed the hook effect [51]. However, no such effect is observed in cultured cells when concentrations of RGC have been extended on the higher side of the IC_50_. Thus, the IC_50_ for RGC in rat and human cell lines has been estimated from more concentrations above than below the IC_50_ (Figure 2C and Figure 3D,H) (Appendix A) which may reflect that the SEs are relatively high at 30–40% of the means, at least in the human cell line (Figure 3D,H) (Appendix A). Further studies on the half-lives of RGC in mice and its effect on mouse 4T1 cells in vitro are needed to determine whether such a hook effect may be occurring here. In contrast, the immunohistochemical staining for S100A4 of the very occasional experimentally produced lung metastasis shows no significant decline with RGC (Table 2) and suggests that those lung metastases which survive do so because of a failure of RGC to trigger destruction of S100A4. The switch to a predominantly nuclear location for S100A4 in primary tumours in mice treated with RGC may suggest changes in S100A4 modification/transport [52,53], but its biological significance here is unknown (Appendix A). US-10113 produces no significant reduction in the number of mice with experimentally induced metastases which suggests that its level of 13.5 mg/kg (about 50 µM) is insufficiently above its IC_50_ in cultured cells (46–56 µM) to produce a biological effect.

The inhibitory results for RGC are independent of the model system used, whether they originated from a rat, mouse or human TNBC source, therefore demonstrating their universality despite the slightly different protein sequences of the respective S100A4s [9,33]. Where the sequence and potential regulation of S100A4 may be important, knockdown experiments have been conducted in the human cell line model as it is closest to the clinical situation. There was little effect of RGC on cell proliferation of the MDA-MB-231 cells despite substantial degradation of S100A4 (Figure 3B,D) (Appendix A) and on the number of mice bearing 4T1 primary breast tumours despite inhibiting completely the formation of lung metastases (Table 1). These results are consistent with those observed previously, where S100A4 failed to stimulate cell proliferation and tumorigenesis despite stimulating cell migration/invasion and metastasis of S100A4-overproducing mouse breast cells [29]. Moreover, there was a lack of correlation between immunohistochemical staining for S100A4 and cell proliferation measured by staining for Ki67 in primary breast carcinomas, despite a strong correlation between staining for S100A4 and patient demise from metastatic breast cancer [10,34]. These results taken together establish the sharp contrast between oncogenes like neu and ras which stimulate cell proliferation and tumour formation [15,16] and metastasis-inducing proteins like S100A4 which fail to stimulate DNA synthesis and tumorigenicity but increase cell migration/invasion and metastasis in vivo [29]. In the past, there have been several reports of the use of PROTACs to demonstrate anti-tumour activity, usually in cMYC-dependent, rapidly growing, cell-xenografted tumours in immunodeficient mice, e.g., mantle cell lymphoma [54] and gastric cancer [55]. These model systems rely on PROTAC targeting bromodomain-containing 4 (BRD4) which, as a powerful transcriptional activator, can increase general transcription and, in particular, that of MYC [56,57,58]. More recently, such an approach has been applied to TNBC [59] and, with monoclonal antibodies, to HER2 positive [60] cell-xenografted, immunodeficient mice. Moreover, when PROTACs have been synthesised with aptamers [61] or double-stranded oligonucleotides that recognise DNA-binding proteins [62], the efficiency of degradation is increased still further. However, these model systems have mainly been established to test the antitumour effects of suppression of transcription factor activity for the production of oncogenes such as cMYC and not the effect on proteins like S100A4 which promote metastasis directly [15,16].

Previous work with inhibitors of S100A4 and the closely related S100P fall into the category of small molecular weight inhibitors of transcription such as sorafenib in human osteosarcoma cells [22], calcimycin [24] and the antihelminth niclosamide [23] in model colon cancer systems, all of which directly inhibit Wnt/β-catenin and thereby S100A4-transcription [23,24] or viral-mediated RNA interference in colon cancer cells [63]. The small molecular weight inhibitors are not very specific and inhibit transcription of other important genes, whilst it is very difficult to direct interfering RNA to the appropriate cells in vivo. Another approach has been to inhibit the interaction of S100A4/P with its reported cell surface target, the receptor for advanced glycation end products (RAGE), either using monoclonal-antibody-targeting [64] or antagonistic peptides [65]. However, their effects are relatively modest and require very high concentrations, presumably because the RAGE receptor, at least in our rat mammary systems, makes only a small contribution to cell migration [50]. Two other approaches to identify small molecular weight inhibitors of the S100 proteins have been reported: the first based on blocking the S100B-p53 tumour suppressor interaction [27] and the second to oligomerise them [66]. The first approach yields compounds based on pentamidine, some of which inhibit the growth of primary malignant melanoma cells in culture in the 100 µM range [67]. The second yields compounds based on phenothiazines, and these can sequester the S100 proteins and thereby indirectly prevent the S100-mediated disassembly of myosin NMIIA, once again in the 100 µM range [66]. However, no direct inhibitors of this interaction have been reported to our knowledge.

Possible limitations to the use of RGC as an antimetastatic agent fall into two categories, those of the target and those of the drug. Firstly, cell migration, even in MDA-MB-231 cells, is not reduced completely by RGC (10–24% remaining) which suggests that another protein, probably S100P, needs to be targeted by a similar but specific drug to inhibit migration almost completely (Figure 4D,I). Secondly, PROTAC RGC has comparatively high molecular weight for a conventional therapeutic drug, and it may therefore suffer from relatively poor solubility, cell permeability, and stability, particularly in vivo [68]. However, PROTAC technology is evolving sufficiently fast so as to ameliorate some of these potential problems in the future [69]. Here, our PROTAC RGC exhibits no toxicity up to a concentration of at least 100 µM in the parental Rama 37 cell line lacking S100A4, but it is biologically active in inhibiting cell migration/invasion at about 10 nM in cell lines containing S100A4 (Figure 2B,F,G and Figure 3H); this yields a potential therapeutic index in vitro of at least 10,000-fold. Consistent with its low in vitro toxicity, RGC produces no obvious in vivo pathologies in mice, despite its substantial reduction of S100A4 in host tissues (Table 2), in agreement with the results for S100A4 gene deletion in transgenic mice [70,71]. Moreover, since preliminary experiments have shown that when 0.07 mg/kg (about 100 nM) RGC is injected two weeks after the formation of experimental metastases in the mouse lungs, there is a significant fall in their number (Appendix A); RGC may be able to eliminate metastases as well as prevent their formation. In conclusion, we have shown that destructive targeting of the protein S100A4, often associated with accelerated deaths of patients [10,12], provides the first realistic chemotherapeutic approach to selectively inhibiting metastasis in humans.

## Figures and Tables

**Figure 1 biomolecules-13-01099-f001:**
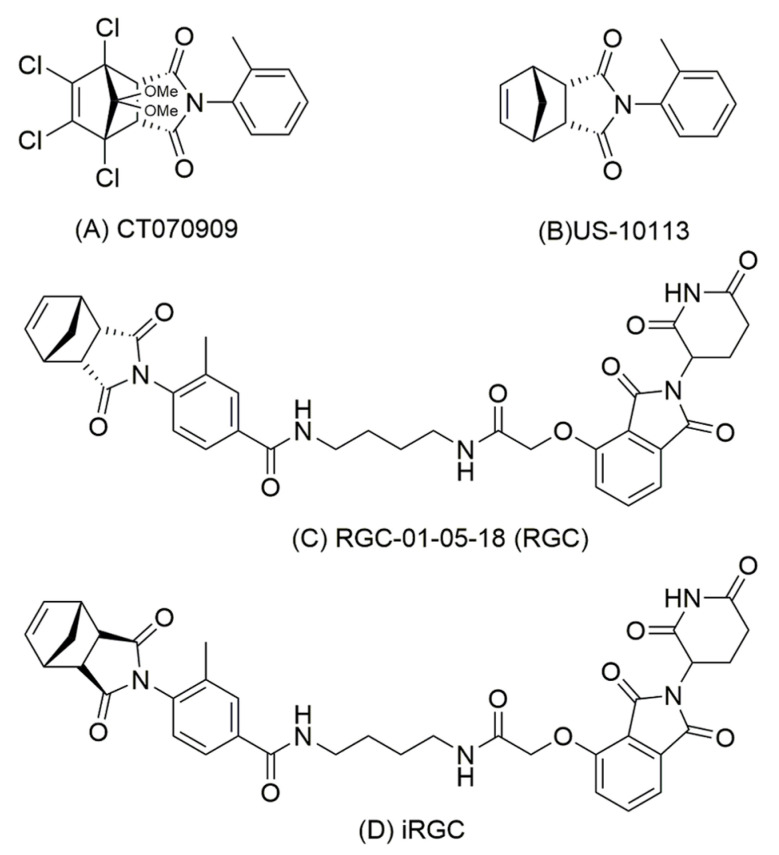
**Chemical structures of compounds.** (**A**) CT7070909 obtained from screening compound library. (**B**) US-10113 synthesised in house. (**C**) The endo isomer PROTAC RGC-01-05-18 showing thalidomide coupled through a linker 4-aminobutylacetamide to US-10113. (**D**) Exo isomer PROTAC iRGC.

**Figure 2 biomolecules-13-01099-f002:**
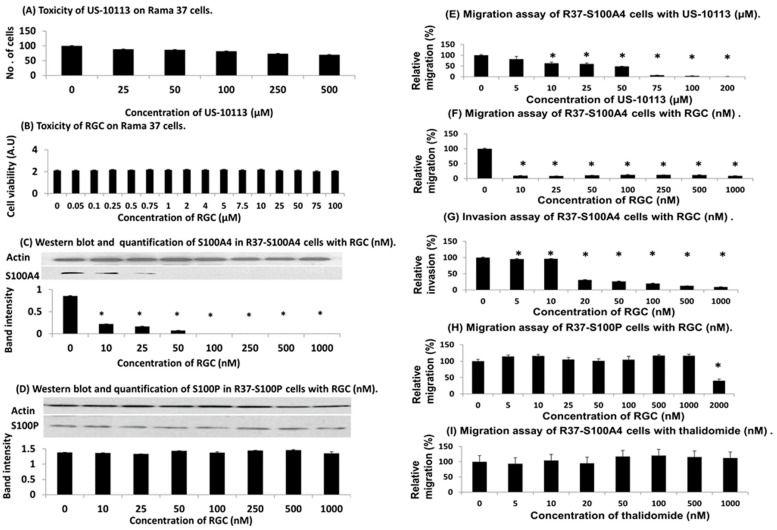
**Effect of inhibitors on rat breast cells.** (**A**) Potential cytotoxicity of US-10113 on Rama 37 parental breast cell line. Cells were grown in different concentrations of US-10113, detached using trypsin/EDTA and counted; results are the mean ± SE of 3 independent experiments normalised to that for no US-10113, standardised to 100 cells. (**B**) Potential cytotoxicity of RGC on Rama 37 cells using sulphorhodamine B assay (SRB). Cultures were fixed in trichloroacetic acid, the residue-bound SRB solubilised, and the resultant optical density measured in arbitrary units (A.U). Results are the mean ± SE of 3 independent experiments, details found in Appendix A. (**C**,**D**) Western blots and quantifications of (**C**) S100A4 and (**D**) S100P in cell lines treated with RGC. Procedures are described in Materials and Methods. The ratio of the scanned areas under the peak band intensities of the S100A4 and S100P blots to that of actin were recorded and the means of the ratios ± SE of 3 independent experiments plus one representative blot are shown. Full-length blots are shown in Appendix A, and numerical values recorded in Appendix A (**E**–**I**). Migration and invasion assays. (**E**,**F**,**H**,**I**) Transwell migration or (**G**) invasion assays were carried out with either (**E**–**G**,**I**) S100A4-overexpressing Rama 37 cells or (**H**) S100P-overexpressing Rama 37 cells for 24 h as described in Appendix A. The average number of cells that migrated in 4 wells in 24 h normalised for the number of cells in the no compound control wells was recorded as a percentage, and the overall mean percentage ± SE for 3 independent experiments is shown. Asterisks indicate difference between zero and the given concentration of inhibitor is significant in Student’s *t* test (*p* < 0.05).

**Figure 3 biomolecules-13-01099-f003:**
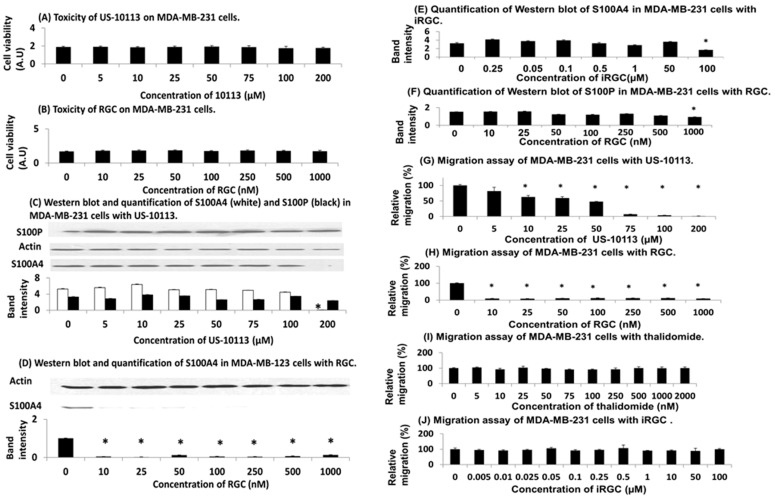
**Effect of inhibitors on human TNBC cells.** (**A**,**B**) Potential cytotoxicity of (**A**) US-10113 and (**B**) RGC on MDA-MB-231 cell line using SRB assay. The resultant optical density of the solubilised SRB protein extract was measured in arbitrary units (A.U) as described in Appendix A. Results are the mean ± SE of 3 independent experiments. (**C**–**F**) Western blots and quantification of S100 protein levels in MDA-MB-231 cells treated with inhibitors. The ratio of the band intensities of S100A4 and/or S100P to actin was recorded (Materials and Methods) and the means ± SE of 3 independent experiments are shown for the following: (**C**) S100A4 and S100P in cells treated with US-10113. Full-length blots are shown in Appendix A and numerical values recorded in Appendix A. (**D**) S100A4 in cells treated with RGC. Full-length blots are shown in Appendix A and numerical values recorded in Appendix A. (**E**) S100A4 in cells treated with isomer iRGC. (**F**) S100P in cells treated with RGC. (**G**–**J**) Effect of (**G**) US-10113, (**H**) RGC, (**I**) thalidomide and (**J**) iRGC on migration of MDA-MB-231. Details are described in Appendix A. Each experiment was conducted 3 times and the overall mean ± SE is shown as a percentage of cells migrating in 24 h relative to that with no compound which was set at 100%. Asterisks indicate difference between zero and the given concentration of inhibitor is significant in Student’s *t* test (*p* < 0.05).

**Figure 4 biomolecules-13-01099-f004:**
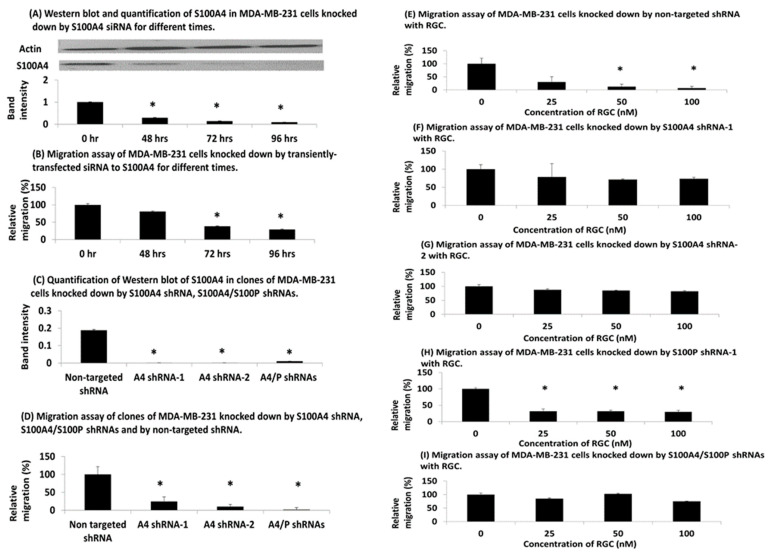
**Knockdown of S100A4 in TNBC MDA-MB-231 cells.** (**A**,**B**) Transiently transfected and depleted for S100A4 showing (**A**) Western blot and (**B**) cell migration of resultant MDA-MB-231 cells. Duplicate knockdowns were initiated and at different times experiments were terminated for Western blots for S100A4 or for measurement of cell migration in 24 h as described in Appendix A. The mean ± SE of 3 independent experiments is shown either (**A**) as ratios of the band intensities of S100A4 to actin relative to that at transfection time zero (representative full-length blots are shown in Appendix A and all numerically recorded in Appendix A) or (**B**) as a percentage relative to the migration rate at transfection time zero (relative migration). (**C**–**I**) Permanently transfected and depleted for S100 proteins showing (**C**) Western blots for S100A4 (numerically recorded in Appendix A) and (**D**–**I**) migration of resultant MDA-MB-231 cell lines in 24 h. (**C**,**D**) Permanently shRNA scrambled nontargeted shRNA (nontarget shRNA), knocked down cell lines for S100A4, S100A4 shRNA clone 1 (A4 shRNA-1), S100A4 shRNA clone 2 (A4 shRNA-2), and for S100A4 and S100P, S100A4/S100P shRNA were subjected to (**C**) Western blots or (**D**) 24 h migration assays as described in Appendix A. The ratios of band intensities of S100A4 to actin in **C** and migration in 24 h normalised as a percentage to that for nontarget shRNA (relative migration), the latter set at 100%, in (**D**) are the means ± SE of 3 independent experiments. (**E**–**I**) Migration of shRNA transfected cell lines for (**E**) nontarget shRNA, (**F**) S100A4 shRNA-1, (**G**) S100A4 shRNA-2, (**H**) S100P shRNA, and (**I**) S100A4/S100P shRNA cells with different concentrations of compounds. The ratio in percentage of migrating cells in 24 h relative to that without compound (100%) is shown as the mean ± SE of 3 independent experiments. There was a significant drop in migrating cells for 25 nM RGC for nontargeted shRNA (Student’s *t*-test, *p* = 0.0028), for S100P shRNA (*p* = 0.0021) but not for S100A4 shRNA-1 (*p* = 0.23), S100A4 shRNA-2 (*p* = 0.22) and S100A4/S100P shRNA (*p* = 0.41). Asterisks indicate difference between zero and the given concentration of inhibitor is significant in Student’s *t* test (*p* < 0.05).

**Table 1 biomolecules-13-01099-t001:** Effect of inhibitors on invasion and metastasis in TNBC mouse model in vivo.

Route of Administration ^a^	Inhibitor	Dose (mg/kg)	Invasion ^b^ Incidence	*p* ^c^	Metastasis ^d^ Incidence	*p* ^c^
Intravenous (iv)	None	-	n.a.		5 of 5	-
	thalidomide	1.35	n.a.		3 of 5	0.22
	US10113	13.5	n.a.		4 of 5	0.50
	RGC	0.07	n.a.		1 of 5	**0.024**
	RGC	0.70	n.a.		1 of 5	**0.024**
	RGC	3.50	n.a.		2 of 5	0.083
Mammary fat pad (sc)	None	-	8 of 8	-	8 of 8	-
	RGC	0.007	0 of 6 ^e^	**0.0003**	0 of 7	**0.0002**
	RGC	0.07	1 of 7	**0.0014**	0 of 7	**0.0002**
	RGC	0.70	3 of 7 ^e^	**0.026**	0 of 7	**0.0002**
	RGC	3.50	2 of 8	**0.0035**	4 of 8	**0.038**

^a^ Mouse TNBC 4T1 cells were injected intravenously (iv) into tail vein (10,000 cells) or orthotopically into mammary fat pad (sc) (7500 cells) of female 4-week-old Balb/c syngeneic mice and the drug was injected (sc) on the same day and then every 2 to 3 days. Mice were sacrificed and autopsied after total of 3 or 4 weeks, respectively. ^b^ Invasion incidence was the number with muscle invasion/number of mice with tumours determined from H&E-stained histological sections of the primary tumour (Materials and Methods), n.a. = not applicable. ^c^ Probability (*p*) of the difference from no inhibitor was calculated using Fisher’s Exact test, 2-sided, *p* < 0.05 was considered significant indicated **in bold.**
^d^ Metastasis incidence was the number with experimental lung metastasis/number of mice for iv route or number with lung metastases/number of mice with tumours for sc route, determined from H&E-stained histological sections of the lungs (Materials and Methods). ^e^ No significant difference in invasion between 0.007 and 0.70 mg/kg RGC, *p* = 0.19.

**Table 2 biomolecules-13-01099-t002:** Immunohistochemical staining for S100A4 in inhibitor-treated mice.

Route of Administration	Inhibitor (mg/kg)	Tissue Staining ^a^		
		Spleen	Colon	Tumour
		Percent ±SD (n) (*p*) ^b^	Percent ±SD (n) (*p*) ^b^	Percent ±SD (n) (*p*) ^b^
Intravenouscells (iv)	Untreated	32.5 ± 8.7 (4)	37.5 ± 8.7(4)	40 ± 18(5)
	RGC (0.07)	17.0 ± 4.5(5) **(*p* = 0.01)**	12.5 ± 6.1(5) **(*p* = 0.0014)**	17.5 ± 0(1) (*p* = 0.32)
	RGC (0.7)	10.5 ± 2.7(5) **(*p* = 0.001)**	10.0 ± 0.0(5) **(*p* = 0.008)**	50 ± 0(1) (*p* = 0.64)
	RGC (3.5)	13.7 ± 7.5(4) **(*p* = 0.017)**	17.5 ± 10.4(4) **(*p* = 0.026)**	45 ± 0(1) ^d^ (*p* = 0.81)
	US10113 (13.5)	26.2 ± 8.5(4) (*p* = 0.34)	40.0 ± 10.8(4) (*p* = 0.73)	42 ± 15(4) (*p* = 0.83)
	Thalidomide (1.35)	26.5 ± 13.2(4) (*p* = 0.46)	37.0 ± 4.5(5) (*p* = 0.91)	35 ± 21(3) (*p* = 0.76)
Cells s/c into mammary fat pad	Untreated RGC (0.007)	23.6 ± 3.8(7) 6.7 ± 1.8(7) ^c^ **(*p* < 0.0001**)	39.3 ± 10.2(7) 11.8 ± 3.4(7) **(*p* = 0.0002)**	63.6 ± 18.2(7) ^e^ 6.1 ± 2.0(7) **(*p* = 0.0001)**
	RGC (0.07)	9.9 ± 2.6(7) **(*p* < 0.0001**)	17.1 ± 9.1(7) **(*p* = 0.001)**	13.5 ± 18.6(7) **(*p* = 0.0003)**
	RGC(0.7)	11.4 ± 3.1(6)^c^ **(*p* < 0.0001)**	13.3 ± 1.4(7) **(*p* = 0.0005)**	14.0 ± 3.2(7) **(*p* = 0.0003)**
	RGC(3.5)	12.5 ± 5.0(8) **(*p* = 0.0004)**	16.9 ± 10.7(8) **(*p* = 0.0012)**	29.6 ± 30.2(7) ^d^ **(*p* = 0.026)**

^a^ Immunohistochemical staining of parenchymal (spleen, colon) or carcinoma (tumour) cells was shown as a percentage ± SD (n= no of tissues or tumours analysed) after incubation of histological sections obtained from tissues in Table 1 with antibodies to S100A4 (Thermo Scientific), diluted 1/200 and incubated for 70 min at room temperature (Appendix A). ^b^ Probability of difference from untreated mice was calculated using Student’s *t*-test (2 sided), *p* < 0.05 was considered significant indicated **in bold.**
^c^ Significant difference was seen between the two, *p* = 0.006. ^d^ One mouse tumour from each set was too small to stain and ^e^ one mouse died prematurely.

## Data Availability

The data sets and materials produced during the present study are available from the corresponding authors upon reasonable request.

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
