# Peer review of "Targeted Destruction of S100A4 Inhibits Metastasis of Triple Negative Breast Cancer Cells"

_biomolecules, 2023, doi:10.3390/biom13071099_

Round 1

Reviewer 1 Report

In this study the authors identified an effective inhibitor that induced degradation of the calcium binding protein S100A4 and concurrently reduced the migration of several breast cancer cells in vitro and breast cancer metastasis in animal models. While the production of a PROTAC (proteolysis targeting chimera) the ‘RGC’ in which the small molecule inhibitor of S100A4, US-10113 was attached to thalidomide, result in effective degradation of S100A4, the other aspects of the studies require further investigation to support the proposed conclusion “In conclusion, we have established proof of principle that destructive targeting of S100A4 provides the first realistic chemotherapeutic approach to selectively inhibiting metastasis”

Major comments:

(1)  Model systems: the authors start with Rama 37 cells (Rat cells) and overexpressed S100A4 or S100P.  It is unclear what is the basal level of these genes/proteins,  what is the level of their overexpression (qPCR, WB), and how their overexpression enhanced migration compared to the parental cells. The authors then characterized the effect of S100A4 knockdown/RGC in human MDA-MB-231 in vitro, and then on murine 4T1 in vivo, without any molecular characterization of these cells in vitro.

(2)  The results are presented mainly by bar graphs without any real imaging of the performed assays except a few WB.

(3)    The in vivo studies were performed only on a few samples (n=5 for IV), without any supporting images. The only available data is the H&E staining shown in the Suppl information, and there the results are also not clear in the present form.

(4)  Throughout the manuscript there are inaccurate terminology, conclusions, scientific terms as well as inconsistent information.   

(1)  The author didn't provide any mechanistic data that can explain the role of S100A4 in breast cancer migration and metastasis. 

There are too many issues related to English and the style, starting with “TNBC” rather than “TRNBC”, IC50 instead of “IC50%,” space between number and molarity (for example: 3.2 nM not 3.2nM) and too many others. I suggest a rigorous editing 

Reviewer 2 Report

This is a well written and interesting paper on the use of a PROTAC (proteolysis-targeting chimera) protein as a possible anti-metastatic therapeutic against breast cancer. In this case using a chemically modified inhibitor of S100A4 (US-10113), linking it to thalidomide enabling its rapid targeting for proteasomal destruction. S100A4 is also known inaccurately as fibroblast specific protein 1 (FSP1) in the scientific literature.

   The authors use a variety of TRNBCs (rat cells, human cells and a syngeneic transplantable mouse model) to show that the new compound is much more effective and less cytotoxic than the parent molecule (US-10113).          

It does not seem at all scientific to name the chimeric protein after one of the authors, ‘RGC’. Surely an acronym of the molecule would be more appropriate and informative. Overall this is an excellent investigation.

Minor points

  1. Figs 2A and 2B – why A is no. of cells while B is AU; Figs 3A and B are both in AU
  2. Line 266 – transfecteded
  3. Fig. 4E – in title ‘Knoked’
  4. Table 1 last line of legend ‘e’: 0-6 is 0.00003, 3 of 7 is 0.026, where does P=0.19 come from?

Reviewer 3 Report

The manuscript by T. Ismail et al. describes the effects of metastasis-promoting protein S100A4 inhibition and destruction using PROTAC. In general, it is a well-designed and described study. I have just several minor comments for its improvement:

Line 88 – check the vector name (https://www.novoprolabs.com/vector/V11029);

Figures 2, 3, and 4 have too little font to be comfortably readable, especially in the printed version;

Chapter 3.2 –toxicity of US-10113 is compared to its conjugate with thalidomide, RGC. What is the standard toxicity of thalidomide itself for the tested cell lines (if tested)?

Round 2

Reviewer 1 Report

In the revised version, the authors indeed replied to the major points that have been raised. However, most of the answers were relied on previously reported studies, which must be shown in the context of the presented results. Hence, answer related to comment No. 1, and the basal level of protein and transcript level of control cells must be included under similar experimental setting. Likewise, major effects on 4T1 in vitro should be included in the current manuscript (possibly in Supplementary). As for imaging to be included, the authors should show a representative result to transfer the biological context of the findings. Lastly, although the manuscript was edited, still TNBC in the abstract included ‘receptor’.

N/A
